

# Enhancing drone autonomy through cloud integration: a comprehensive software architecture for navigation, visual servoing, and control

Muhammad Bilal Kadri

College of Computer and Information Sciences, Prince Sultan University, Riyadh, Saudi Arabia

## ABSTRACT

This work proposes a comprehensive software framework for cloud-enabled autonomous drone navigation, featuring precise target tracking *via* image-based visual servoing (IBVS) coupled with a control scheme. In this study, a low-cost quadcopter running the ArduPilot firmware is evaluated within a simulation-in-the-loop (SITL) environment using a Gazebo-based simulation of a real-world mission. The tested software architecture can be seamlessly integrated with an onboard companion computer for real-time execution. The mission involves waypoint tracking, precise identification and descent onto visual markers using IBVS, along with real-time data visualization on a remote client connected *via* a cloud interface. Because the software architecture is versatile, it can accommodate any conventional or knowledge-based controller. To demonstrate the efficacy and robustness of the proposed architecture, the quadcopter was tested under challenging weather conditions, where it successfully completed the mission despite disturbances and sensor noise. Finally, the complete software architecture has been tested and implemented in the robot operating system (ROS).

# INTRODUCTION

Uncrewed aerial vehicles (UAVs) are autonomous or remotely piloted aircraft that operate without an onboard human pilot. These versatile flying machines have revolutionized various industries and gained widespread adoption due to their unique capabilities. In today's world, UAVs have a broad range of applications across sectors such as aerial surveillance (*Papachristos et al., 2019*), agriculture (*Radoglou-Grammatikis et al., 2020*), infrastructure inspection (*Ly & Phung, 2020*; *Molina, Huang & Jiang, 2023*), search and rescue operations (*Akhloufi, Couturier & Castro, 2021*; *Khan et al., 2023*), delivery services (*Patrik et al., 2019*), environmental monitoring, and more (*Molina, Huang & Jiang, 2023*).

With a significant shift in research focus toward achieving drone autonomy (*Tanveer & Kadri, 2024*), there is a growing desire to reduce reliance on human operators (*Wang et al., 2024*). The goal is to enable UAVs to perform complete operations without constant human intervention or supervision (*Khalil et al., 2022*), thereby increasing productivity, reducing costs, and improving safety (*Youn et al., 2021*). Over time, numerous

Corresponding author
Muhammad Bilal Kadri,
mkadri@psu.edu.sa

techniques have been developed to achieve drone autonomy, including sensor fusion (*Abdelkader et al., 2025*; *Yousuf & Kadri, 2025*), state estimation, computer vision (*Jitoko et al., 2021*), and navigation (*Bijjahalli, Sabatini & Gardi, 2020*; *Patrik et al., 2019*). Nevertheless, the realization of a fully integrated, cloud-enabled software architecture that supports real-time autonomous navigation, robust target identification, and high-precision positioning—while imposing minimal onboard computational overhead (*pjrambo, 2025*)—remains a significant technical challenge (*Gustave, Chahal & Belbachir, 2020*; *Gao et al., 2024*). Few simulation architectures have been proposed (*Louali et al., 2023*) but they are limited to navigation and do not cover target identification and cloud integration.

Traditional methods, such as GPS-based navigation, may not provide the required level of accuracy for UAVs operating over small target areas because the typical GPS positioning tolerance is between 1–6 m (*Pu, Shi & Gu, 2021*). To address this challenge, image-based visual servoing (IBVS) guidance systems offer a promising solution for achieving precise positioning of autonomous UAVs (*Van Kirk et al., 2022*).

Vision-based guidance systems leverage computer vision techniques to extract critical information from a target location (*Jitoko et al., 2021*; *Pluckter & Scherer, 2020*). By analyzing visual features captured by onboard cameras or sensors, these systems can adjust a UAV's position and orientation relative to its surroundings with high precision (*Qin, Li & Shen, 2018*). This approach enables fine-grained control of the UAV's flight path in real-time, allowing it to navigate complex, dynamic environments with high accuracy, regardless of GPS signal quality (*Singh & Sujit, 2016*). The IBVS (*Fu et al., 2023*) system can also be integrated with a controller to achieve precise positioning and tighter control.

The robot operating system (ROS) provides a flexible and modular framework for building robotic systems, including autonomous UAVs (*Papachristos et al., 2019*; *Honig & Ayanian, 2017*; *Raheel et al., 2024*; *Mehmood et al., 2024*). ROS offers a comprehensive suite of tools, libraries, and communication protocols that facilitate the development of complex autonomous systems. By integrating an IBVS system with MAVROS, developers can leverage the robustness, scalability, and interoperability of ROS to create efficient, reliable autonomous UAV platforms (*Gustave, Chahal & Belbachir, 2020*; *Honig & Ayanian, 2017*; *Meyer et al., 2012*).

In this article, we present a fully integrated software architecture that uses IBVS to enable precise positioning of autonomous UAVs over target areas (*Aoki & Ishigami, 2023*). The IBVS module can be combined with either a conventional or a knowledge-based controller, and the system readily supports cloud integration for remote data logging. By leveraging the ArduPilot firmware (*ArduPilot Development Team, 2025*), the architecture remains broadly compatible and can be easily implemented within the ROS/Gazebo simulation environment. To develop a robust software framework, we utilize a specific mission scenario namely, the flight mission outlined in the Teknofest 2024 International UAV Competition Rule book which serves as a convenient, practical problem scenario for our investigation. Furthermore, we discuss the seamless integration of our software algorithms with ROS, capitalizing on its modular approach to development. By employing

this combined approach, we aim to demonstrate the effectiveness of the proposed software architecture for autonomous UAVs in real-world scenarios.

## Contribution of the proposed work

The novelty of this work lies in the seamless integration of various components into a unified, modular, and extensible software architecture a feature largely missing in existing literature. The design and demonstration of a comprehensive end-to-end system, where real-time control (*via* PID and IBVS), sensor feedback, cloud communication, and remote visualization operate in complete synchrony. This level of interoperability and system coherence, especially under challenging conditions such as wind disturbances and sensor noise, has been achieved through careful architectural planning and validation in a high-fidelity simulation environment. The system supports ROS 2, ensuring compatibility with modern tools and enabling plug-and-play integration of emerging technologies. The updated architecture supports adaptive or AI-based control methods, as well as advanced cloud platforms such as AWS RoboMaker, Azure Internet of Things (IoT) Hub, and Google Cloud Platform. This makes the system future-proof, highly adaptable, and directly usable for a broad range of UAV applications. To the best of our knowledge, very few works offer this level of architectural completeness, cross-domain integration, and extensibility, which makes the contribution both novel and impactful within the field of autonomous UAV systems.

In summary, this work presents a unified, modular, and cohesive (integrated) control framework that can be feasibly deployed on low-cost (affordable) UAV platforms for enhancing UAV autonomy in various applications such as environmental monitoring, infrastructure inspection or emergency response. The key novelties of this work include:

- A hybrid UAV control pipeline that combines autonomous waypoint navigation with IBVS for high-precision positioning over dynamic, unknown or GPS-denied targets.
- A cloud-integrated architecture that supports real-time monitoring and remote control, enabling seamless operation of UAVs over the internet.
- A mission-ready software framework that significantly reduces human intervention by enabling end-to-end autonomy from mission assignment to task execution validated in high-fidelity software-in-the-loop (SITL) simulation.

## PROBLEM FORMULATION

The objective of this work is to propose a comprehensive software architecture for drone navigation, control, and target tracking, along with seamless cloud integration. We begin by discussing the basic quadcopter model derived from first principles, followed by a presentation of a generic drone control architecture. Since this work employs the ArduPilot framework, the control scheme integrated with ArduPilot and subsequently with IBVS is presented next. Finally, the section concludes with a discussion of cloud integration.

## First principle quadcopter model

A simplified model of the quadcopter can be given as follows. The thrust $u$ produced by the four motors acting on the quadcopter is:

$$u = \sum_{i=1}^{4} f_i. \tag{1}$$

For $i = 1, \ldots, 4$, $f_i$ is the force produced by motor $M_i$. We can assume $f_i = k\omega_i^2$, where $k_i$ is a constant and $\omega_i$ is the angular speed of the $i$th motor. The generalized torques for the roll, pitch and yaw can be defined as

$$\tau = \begin{bmatrix} \tau_\psi \\ \tau_\theta \\ \tau_\phi \end{bmatrix} \triangleq \begin{bmatrix} \sum_{i=1}^{4} \tau_{M_i} \\ (f_2 - f_4)\ell \\ (f_3 - f_1)\ell \end{bmatrix} \tag{2}$$

where $\ell$ is the distance between the motors and the center of gravity, and $\tau_{M_i}$ is the moment produced by motor $M_i$, around the center of gravity of the aircraft.

## Generic two-layered control architecture

To effectively control the quadcopter, a two-layer control strategy is required. The inner control loop manages altitude, while the outer control loop is responsible for trajectory tracking. The general control scheme is illustrated in Fig. 1. Quadcopters are underactuated systems, with four actuators controlling six degrees of freedom in 3D space. Motion in any direction is achieved through a complex combination of roll, pitch, and yaw maneuvers. Given that the quadcopter is a highly nonlinear and inherently unstable system, various controllers from simple PID loops to complex nonlinear model predictive controllers (MPC) have been proposed in the literature.

## Mathematical formulation for vision-based drone target tracking with PID control

A UAV equipped with a downward-facing camera detects a target at pixel coordinates $(x_{\text{target}}, y_{\text{target}})$. The UAV switches from waypoint navigation to target tracking upon detection of a specific visual marker (BLUE or RED). It uses three PID controllers: $\text{PID}_x$, $\text{PID}_y$, and $\text{PID}_z$, which control motion in the $x$, $y$, and $z$ directions, respectively.

### *Coordinate definitions*

Let:

- $(x_d(t), y_d(t), z_d(t))$: Drone position at time $t$
- $(x_t, y_t)$: Target's position in world frame

The Euclidean distance in the XY-plane is given by:

$$d(t) = \sqrt{(x_t - x_d(t))^2 + (y_t - y_d(t))^2}. \tag{3}$$

Let $\varepsilon > 0$ be the proximity threshold.

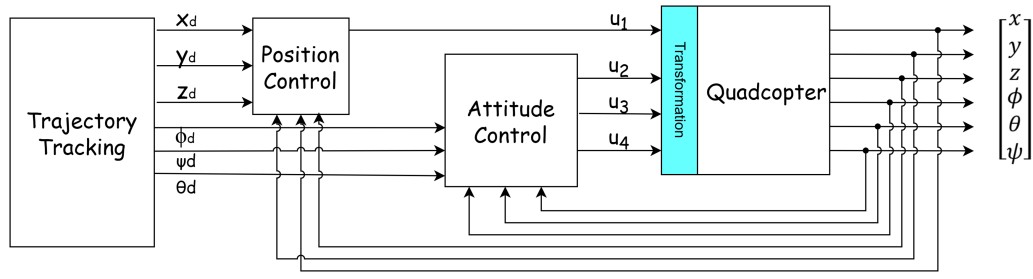

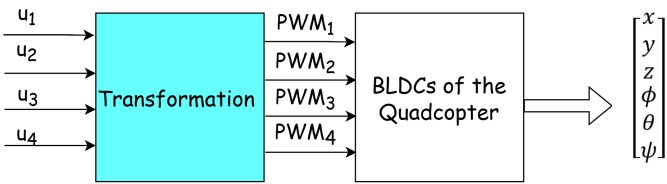

**Figure 1 Generic control architecture for a quadcopter.** The trajectory tracking block outputs the desired positions $(x_d, y_d, z_d)$ and attitudes $(\theta_d, \phi_d, \psi_d)$ which converted into control outputs $(u_1, u_2, u_3, u_4)$. These outputs then undergo a series of hardware-specific transformations in order to generate the signals required to control the four motors of the quadcopter.

## Mode switching

Define the mode $\mathcal{M}(t)$ as:

$$\mathcal{M}(t) = \begin{cases} \text{Waypoint Following} & \text{if no target detected} \\ \text{Target Tracking} & \text{if BLUE or RED target detected} \end{cases}. \tag{4}$$

## PID control laws

*Lateral PID controllers (x and y)*

For $d(t) > \varepsilon$, the drone moves laterally toward the target:

$$e_x(t) = x_t - x_d(t) \tag{5}$$
$$e_y(t) = y_t - y_d(t) \tag{6}$$

$$u_x(t) = K_{p,x} e_x(t) + K_{i,x} \int_0^t e_x(\tau) d\tau + K_{d,x} \frac{de_x(t)}{dt} \tag{7}$$

$$u_y(t) = K_{p,y} e_y(t) + K_{i,y} \int_0^t e_y(\tau) d\tau + K_{d,y} \frac{de_y(t)}{dt}. \tag{8}$$

*Descent controller (z-axis)*

When $d(t) \leq \varepsilon$, lateral motion stops and descent begins:

$$u_x(t) = u_y(t) = 0 \tag{9}$$
$$e_z(t) = z_{\text{target}} - z_d(t) \tag{10}$$

$$u_z(t) = K_{p,z} e_z(t) + K_{i,z} \int_0^t e_z(\tau) d\tau + K_{d,z} \frac{de_z(t)}{dt}. \tag{11}$$

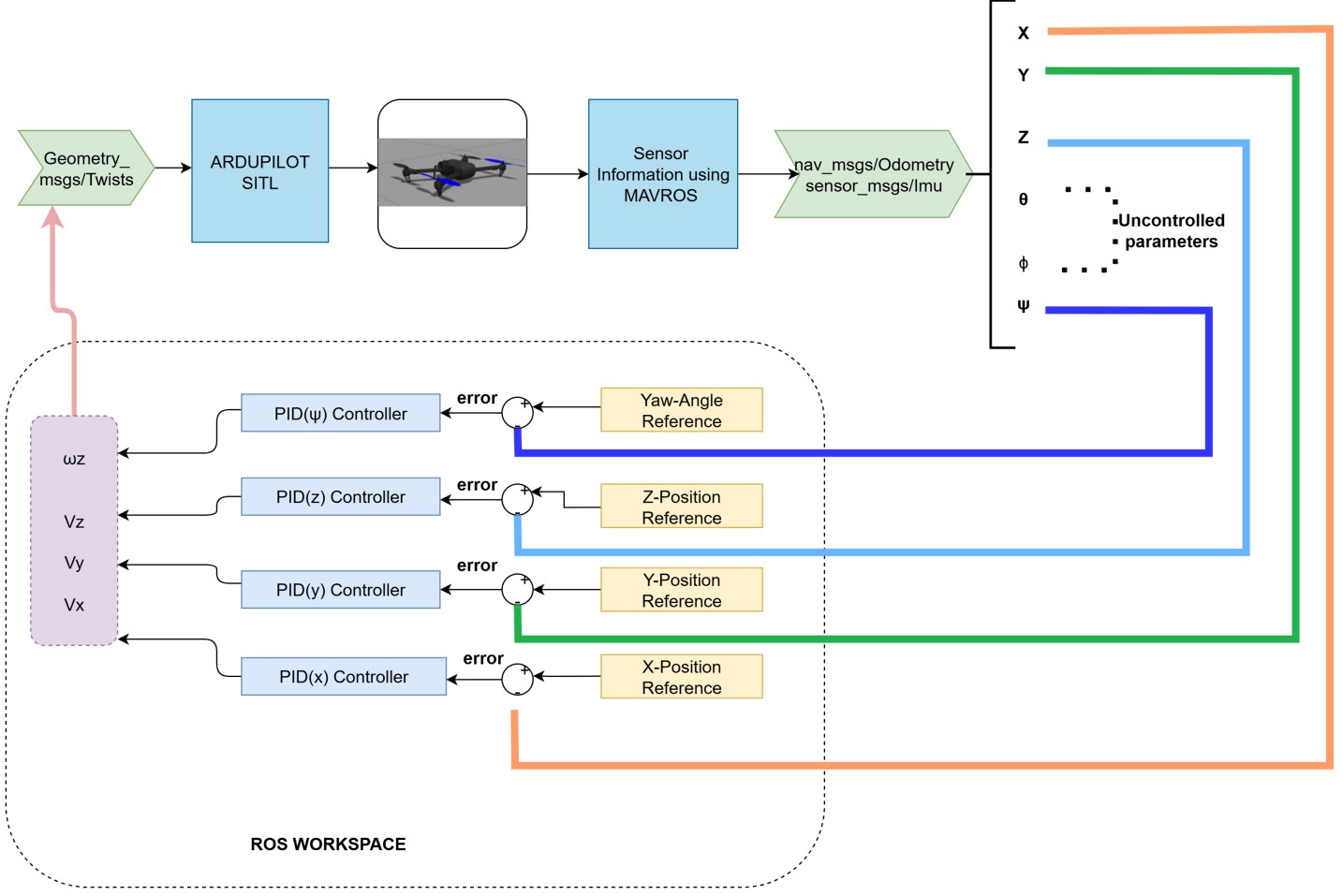

**Figure 2 Control architecture for the proposed software system.** Includes Gazebo IRIS drone model (from ArduPilot Gazebo plugin, licensed under LGPL-3.0).

### Complete control logic

$$
\begin{cases}
u_x(t), u_y(t) \neq 0, \ u_z(t) = 0 & \text{if } \mathscr{M}(t) = \text{Target Tracking and } d(t) > \varepsilon \\
u_x(t) = u_y(t) = 0, \ u_z(t) \neq 0 & \text{if } \mathscr{M}(t) = \text{Target Tracking and } d(t) \leq \varepsilon \, . \\
\text{Waypoint navigation control} & \text{if } \mathscr{M}(t) = \text{Waypoint Following}
\end{cases}
\tag{12}
$$

### Control scheme integrated with ArduPilot firmware

In this study, we use the ArduPilot firmware in a software-in-the-loop (SITL) setting with a Gazebo plugin for the quadcopter. All communication with the quadcopter is handled *via* MAVROS messages. Using the MAVROS interface, the quadcopters (x, y, z) position is controlled by publishing messages on the MAVROS topic `geometry_msgs/Vector3` under the `linear` field. To effectively maneuver the drone in 3D space without directly controlling angular velocities, we designed three independent PID-controllers one for each of the *x*, *y* and *z* axes. The control scheme employing these three independent PID-control loops is shown in Fig. 2.

The use of ArduPilot in a software-in-the-loop (SITL) configuration serves as a robust proof of concept, allowing for accurate validation of the control architecture and seamless transition of the developed code to a companion computer onboard a physical UAV. The SITL environment provides high-fidelity emulation of flight controller behavior, sensor feedback, and MAVLink communication, making it a reliable precursor to HIL and field validation.

## Discussion on different control strategies suitable for the mission

In this study, we adopted a PID-based control architecture due to its computational simplicity and proven reliability in real-time UAV applications. Implementing more complex control strategies such as model predictive control (MPC) or adaptive controllers, while academically appealing, introduces significant computational overhead. This is particularly critical for onboard embedded systems where processing resources are limited, and any increase in computational demand can adversely impact flight time and overall system responsiveness. Moreover, adaptive control methods often rely on online learning or parameter adaptation, which can be risky in mission-critical scenarios. In such cases, transient inaccuracies or model misidentification may result in control instability or false positives during target tracking. In contrast, the use of IBVS in conjunction with well-tuned PID controllers provides a robust and computationally efficient solution that has demonstrated consistent performance in our simulated experiments. MPC-based strategies also require accurate nonlinear system models and finely tuned optimization parameters. These models must be identified and validated prior to deployment, which can be time-consuming and sensitive to environmental variations. On the other hand, the PID controllers used in our work leverage ArduPilot's auto-tuning mechanisms, allowing them to adaptively stabilize the UAV and maintain robust tracking performance across varying environmental conditions, as evidenced in the results presented in the manuscript. Therefore, while we recognize the merits of alternative control strategies, the selected approach strikes a practical balance between performance, robustness, and real-time feasibility for the mission scenarios considered in this work.

## Image based visual servoing

The drone is equipped with a downward-facing camera. Once the target is identified, the controllers in the $xy$-plane are activated to direct the drone toward the target. The drone transitions from waypoint tracking to target tracking mode, aligning itself at the center of the target area before descending through activation of the $z$-controller. The image processing and controller modules work in close coordination to achieve precise positioning, despite sensor inaccuracies or environmental disturbances such as wind gusts. An example of the drone over the target is shown in Fig. 3.

The `Water_Reservoir_Discharge_Location_Identification_CallBack` function is responsible for real-time image processing of the video feed captured by a downward-facing camera mounted on a drone. As the drone flies over the field, the function processes incoming ROS image messages using OpenCV to identify two specific colored targets: a blue circle indicating the water reservoir and a red circle indicating the

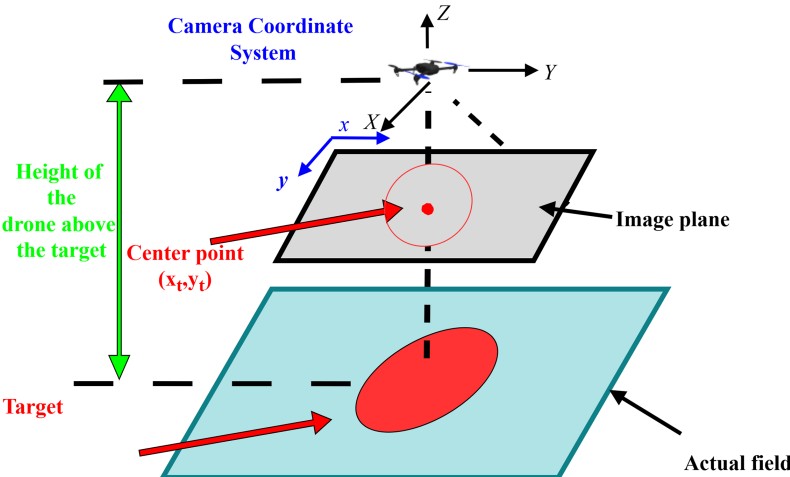

**Figure 3 Camera coordinate system (*Fu et al., 2023*) includes Gazebo IRIS drone model (from ArduPilot Gazebo plugin, licensed under LGPL-3.0).**

water discharge location. The image is first blurred and converted to HSV format to apply color masking for red and blue. After noise reduction *via* erosion and dilation, the function extracts contours of the colored regions and calculates their geometric properties such as centroid and radius using the minimum enclosing circle. The identified (x, y) positions and radii are published as ROS messages, which are then consumed by PID controllers to regulate the drone's position. Specifically, PID controllers in the x and y directions use the difference between the image center (current drone position) and the detected target position to align the drone above the target. Once the target is centered within a predefined threshold, the PID(z) controller is activated to lower the drone vertically. The code also includes visualization for monitoring: green lines represent coordinate axes, black lines connect the center of the image to the target, and text annotations show positional offsets. This system ensures accurate visual identification and autonomous positioning of the drone over the water reservoir and discharge points, enabling a closed-loop vision-based servoing strategy for precise aerial operations.

## Software architecture for cloud

Operational data from the drone's companion computer can be transmitted to a cloud instance using services provided by Amazon Web Services (AWS) (*AWS, 2025*), IBM Cloud, or Microsoft Azure (*Azure, 2025*). In our implementation, we utilize an AWS EC2 instance to transmit and store drone data. Consequently, the complete flight path, current drone location, target area coordinates, and other relevant data are securely stored in a cloud-based database, making them available for offline processing. The cloud software architecture is illustrated in Fig. 4. Data on ROS parameter server are transmitted to the cloud which can be used for visualization on a remote web-client.

Cloud integration offers significant advantages in terms of scalability, remote access, and centralized data management which are important for enhancing drone autonomy in real-world operations. However, despite the numerous benefits, several technical and

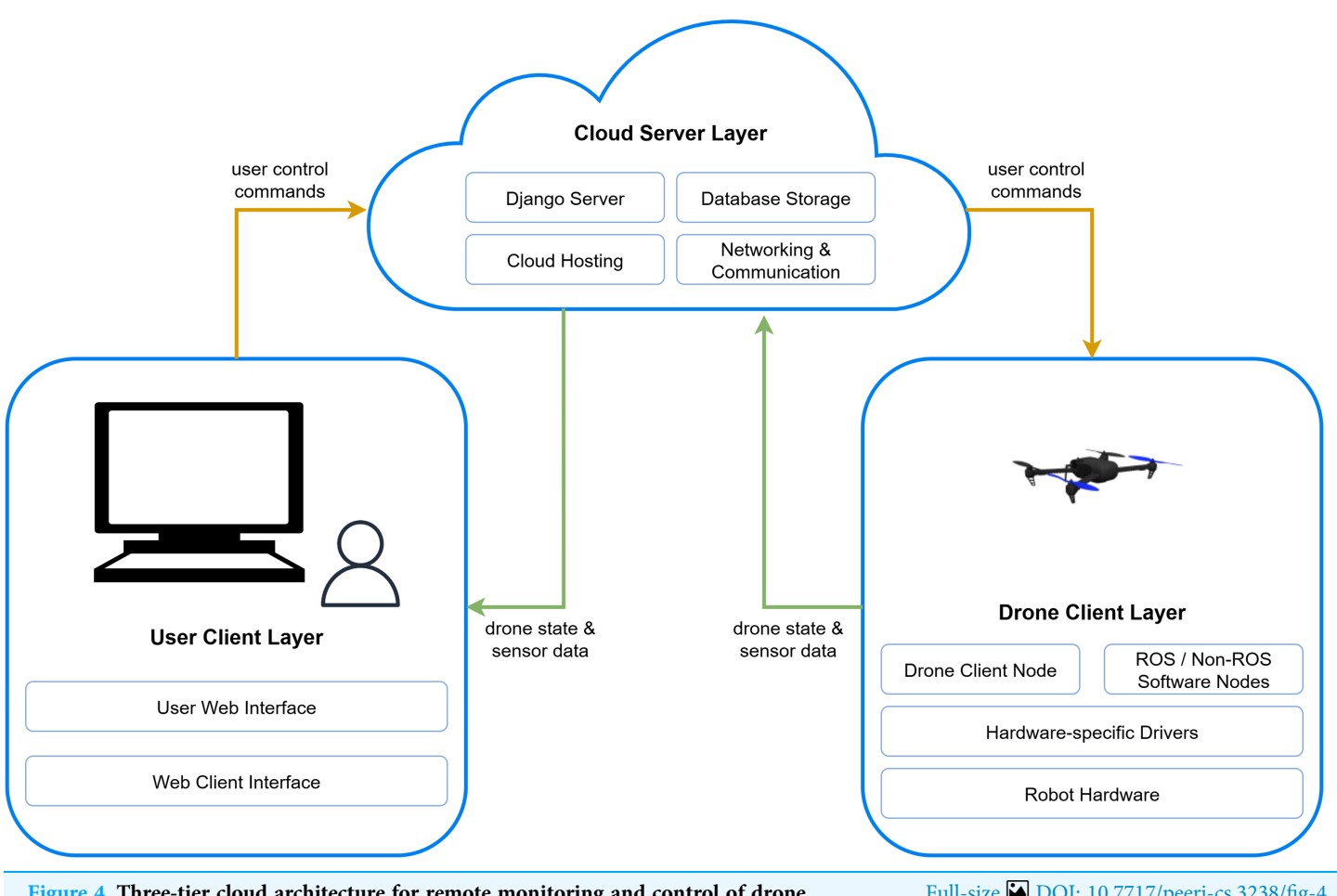

**Figure 4  Three-tier cloud architecture for remote monitoring and control of drone.**

operational challenges related to data latency, system reliability, and security need to be considered.

Since real-time decision-making and control operations are sensitive to communication delays, any latency introduced by network congestion or unreliable connections can significantly impact mission performance. For this reason, our architecture ensures that all high-frequency control loops are executed locally on the drone's onboard companion computer and only non-critical flight data is transmitted over the cloud for remote monitoring. Similarly, the architecture is designed to allow autonomous operations to continue locally in the absence of cloud access in order to improve the system's reliability and fault tolerance during operations in remote environments under adverse network conditions.

Finally, security is one of the most critical concerns when integrating drones with cloud services as transmitting drone data over public networks introduces risks such as unauthorized access, data tampering, and potential system hijacking. The implementation addresses these risks by incorporating robust security measures, including end-to-end encryption and secure authentication protocols, to ensure data integrity and protect against external threats.

## Migration from ROS-1 to ROS-2

The original implementation of our system was developed and validated using ROS 1 (Melodic and Noetic), which was widely adopted at the time. In response to the reviewer's concern regarding the currency of the referenced codebase, we have now updated the associated repository to be fully compatible with ROS 2 Humble *via* the ROS 1—ROS 2 Bridge. This transition to ROS 2 significantly enhances the system's modernity and relevance. ROS 2 offers native support for real-time distributed processing, which is highly beneficial for time-critical UAV operations. It also introduces quality of service (QoS) policies, allowing fine-grained control over communication reliability, latency, and durability—factors that are crucial for UAV control in SITL and cloud-based deployments. Furthermore, ROS 2 supports DDS-Security features such as encryption, authentication, and access control, thereby addressing the cybersecurity requirements of cloud-based UAV operations where data may traverse public or hybrid networks. The ROS 2 update also facilitates seamless integration with modern cloud platforms such as AWS RoboMaker, Azure IoT Hub, and Google Cloud Platform (GCP), making the system more extensible and future-proof. These updates not only modernize the codebase but also strengthen the credibility and practical applicability of the proposed architecture in real-world, mission-critical scenarios.

## MISSION DESCRIPTION

The chosen mission is inspired by the Teknofest 2024 International UAV Competition (*TEKNOFEST, 2024*). The primary theme of the mission is autonomous water transportation. The flight area measures 170 m by 120 m, with a 10 m wide buffer zone surrounding it. Two target locations are placed along the flight path: a blue, circular water intake reservoir with a diameter of 3 m and a height of 70 cm, and a red water release area with a diameter of 2.5 m and a height of 100 cm. The positions of both targets are arbitrary. The UAV is expected to complete two full laps around the flight area following the route defined in Fig. S1. During the first lap, the UAV should identify and record the GPS coordinates of the water intake reservoir and water release area. In the second lap, it should collect and release water at the corresponding target locations. The key features of the mission are summarized as follows:

1. Autonomous takeoff and landing
2. Autonomous waypoint navigation, ensuring that the UAV avoids entering the buffer and forbidden zones
3. Environmental perception to detect target areas during the mission and record their GPS coordinates
4. Identification and localization of the target areas using image processing and/or computer vision
5. Precise positioning of the UAV over the center of the target areas
6. Online visualization of the drone's flight parameters

## Flowchart of the complete mission

The navigation algorithm for the mission is summarized in the three flowcharts included in this section, which illustrate the stages of autonomous takeoff, waypoint following, target identification, IBVS and tracking control, and finally returning to the launch location. These stages are shown in Figs. S2, S3 and S4 respectively.

# SIMULATION SETUP

## Pixhawk

The advent of powerful, compact, and affordable processors has facilitated the development of small, lightweight flight controllers, such as the Pixhawk. These controllers come equipped with built-in inertial measurement units (IMUs) and multiple peripheral interfaces, making it easy to integrate various external sensors and devices. They are compatible with open-source autopilot firmware, such as ArduPilot (*ArduPilot Development Team, 2025*) or PX4 (*PX4 Development Team, 2024*), which provide robust, extensively tested flight stacks to ensure reliable UAV control.

## Software in the loop

SITL (*ArduPilot Development Team, 2024*) is a technique that enables simulation of UAV behavior directly on a development computer, eliminating the need for physical drone hardware. In SITL, the autopilot firmware or flight stack is installed and executed on a local computer or server, thereby removing the requirement for a dedicated flight controller such as the Pixhawk. The firmware communicates with a simulator, such as Gazebo (*Gazebo Development Team, 2024*), by receiving simulated environmental and sensor data from a 3D virtual world and sending corresponding motor and actuator commands to control the drone's position and attitude. SITL is extremely useful for testing various control strategies and debugging code before deployment on real hardware. It allows examination of the drone's response in challenging scenarios that are difficult to replicate in the real world, thereby facilitating the early identification and resolution of persistent issues and reducing the risk of crashes.

One of the core advantages of SITL is that it uses the exact same autopilot firmware intended for hardware deployment which eliminates the need for a physical flight controller during simulation. Our use of ArduPilot and Pixhawk architectures ensures that the firmware and flight logic executed in simulation can be feasibly and reliably deployed on real hardware with minimal or no changes. This ensures that control, estimation, and mission logic behave identically in both environments. Moreover, since SITL shares the same state estimation and control logic as its hardware counterpart, it is widely used in academic and industrial UAV development pipelines as a credible and widely accepted validation stage for algorithms targeting real-world deployment.

## ArduPilot

ArduPilot (*ArduPilot Development Team, 2025*) is an open-source autopilot system that provides plugins and packages for conducting SITL simulations across various platforms (*ArduPilot Development Team, 2024*). The SITL package includes the complete ArduPilot

flight control stack, featuring sensor fusion, state estimation, attitude control, and position controllers. The accompanying plugins serve as interfaces between the SITL packages and the virtual world and drone models within the simulators. This integration allows developers to conduct comprehensive testing of scenarios, such as takeoffs, landings, and navigation which in turn improves the overall performance and reliability of autonomous drones.

## PID autotuning

In order to ensure optimal performance and reduce the manual effort involved in controller tuning, the Autotune feature provided by the ArduPilot SITL environment was utilized. This automated process systematically adjusts the PID gains to achieve stable and responsive control, and the final tuned values were selected based on their consistent performance across multiple test flights. The robustness of these gains is demonstrated in our simulations, particularly under challenging conditions involving high wind disturbances and sensor noise, where the controller maintained stable flight behavior.

## Integrating various modules in the simulation environment

Gazebo leverages 3D physics engines to create virtual environments, or "worlds," that closely resemble the real world in both visualization and underlying physics. Once a world is created in Gazebo, a UAV model is introduced into the simulation. For our purposes, we utilized the 3DR Iris drone model (*ArduCopter, 2024*), which is widely used for simulating MAVLink (*MAVLink Development Team, 2024*) drones. This model has been meticulously designed to accurately represent the drone's physical characteristics, including its dynamics, propulsion system (motors), and sensors. It incorporates various plugins to enhance functionality, such as the Lift Drag Plugin, which calculates the aerodynamic effects of the rotating propellers, and an integrated IMU sensor model to provide necessary sensor data. Additionally, the ArduPilot SITL Gazebo Plugin facilitates communication with the ArduPilot SITL firmware. To simulate a real-world downward-facing camera, an open-source ROS Gazebo Camera Plugin is added to the Iris model; this camera is crucial for identifying and locating target areas.

To test the proposed software architecture (*Qays, Jumaa & Salman, 2020*), the flight area depicted in Fig. S1 was modeled in Gazebo. A basic world was created comprising a ground plane along with the water intake and release target areas. The models for the water intake reservoir and water release area were initially designed using SolidWorks (*Dassault Systèmes, 2024*) and then exported as STL files. These files were used to define the visual and collision geometries of the models. The positions, scales, and orientations of these models were specified in the world file, which can be spawned within the simulation environment.

Finally, several techniques have also been incorporated to reduce the discrepancies between simulation and real-world execution. To ensure high simulation fidelity, the Gazebo simulation has been extended with realistic physical and environmental effects, including wind gusts, lighting variations (shadows) and sensor noise. These enhancements simulate key challenges faced during actual deployments and provide a robust testbed for

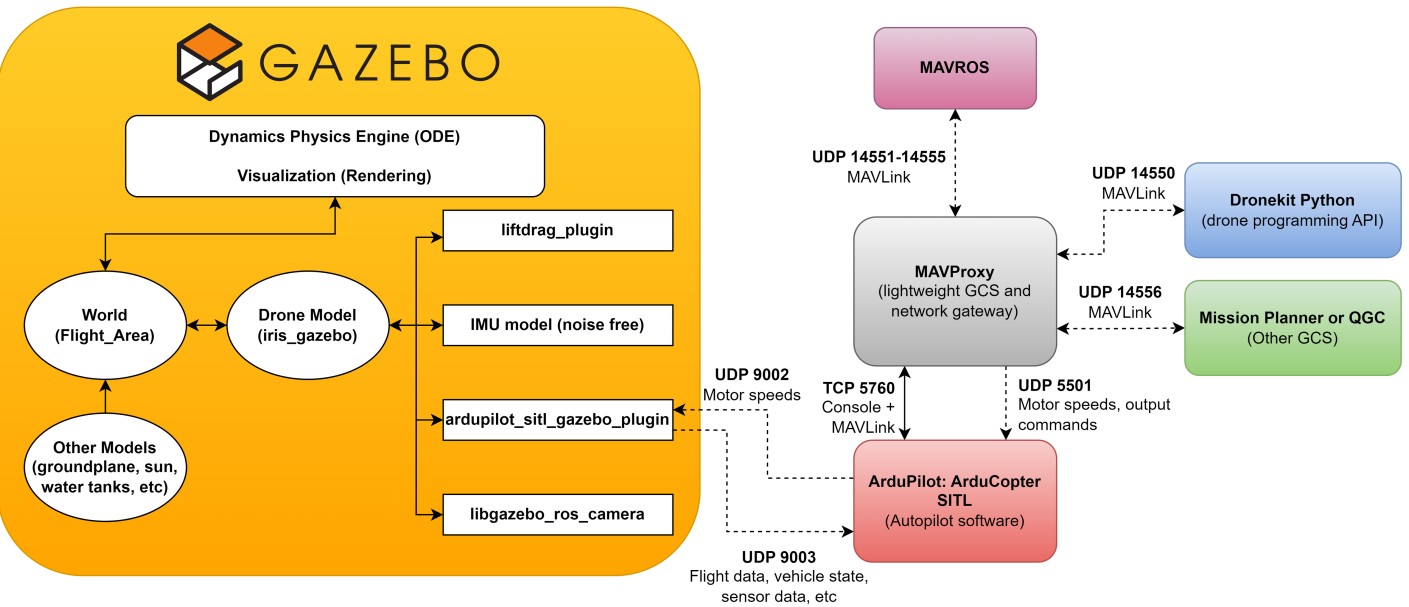

**Figure 5** Autonomous UAV simulation setup using Gazebo, ArduPilot, MAVProxy and MAVROS.

validating control, perception, and decision-making algorithms ensuring confidence that the performance metrics observed in simulation will generalize well to hardware execution.

## Ground control station

Validating SITL functionality is crucial once the simulation environment is configured. A lightweight Ground Control Station (GCS) software, MAVProxy, is used for this purpose. MAVProxy enables the sending of basic MAVLink commands to the UAV model in Gazebo, facilitating operations such as switching to Guided Mode ("mode GUIDED"), arming the drone ("arm throttle"), initiating takeoff to a height of 1 m ("takeoff 1"), and commanding the UAV to land ("mode LAND"). If the behavior of the Iris drone in the simulation aligns with that of a real-world drone, it indicates that the simulation environment has been appropriately configured for code development and testing.

MAVProxy offers several advantages, including network routing capabilities. It can forward MAVLink control commands from other devices or computers on the same network. This feature is valuable when accessing UAV flight information or transmitting control commands from a different GCS that may have a more sophisticated graphical interface, such as Mission Planner or QGroundControl, or from a companion computer utilizing ROS. Figure 5 illustrates how MAVProxy serves as a network gateway, integrating various frameworks such as MAVROS, the DroneKit Python API, and other GCS software like Mission Planner or QGroundControl.

## Potential challenges associated with cloud-based UAV deployments

The peoposed architecture can communicate with ROS 2 topics. Compatibility with ROS 2, which significantly strengthens the system's capability to address real-world concerns

such as security, latency, and reliability in distributed environments. ROS 2 introduces native support for real-time distributed processing and advanced quality of service (QoS) policies, which allow fine-grained control over communication parameters such as message delivery reliability, latency sensitivity, and durability. These features are critical in mitigating data latency and ensuring robust communication for UAV-cloud integration. Moreover, ROS 2's support for DDS-Security extensions enables the implementation of end-to-end encryption, authentication, and access control, directly addressing the cybersecurity concerns associated with cloud-connected UAV operations particularly when sensitive data is transmitted using public or hybrid networks. The updated ROS 2-based architecture also facilitates seamless integration with modern cloud platforms including AWS RoboMaker, Azure IoT Hub, and Google Cloud Platform, each of which provides scalable, secure, and resilient infrastructures for robotic applications. While these capabilities significantly enhance the system's extensibility and readiness for real-world deployment, nevertheless the cloud-based UAV systems inherently face challenges in real-time communications.

# TESTING THE MISSION SCENARIO IN THE GAZEBO WORLD USING ROS FRAMEWORK

The ROS framework is divided into three main modules, each spawned as a separate ROS node to run in parallel. These nodes share information *via* ROS topics or through the ROS Parameter Server. The three primary ROS nodes are as follows:

1. **Autonomous waypoint navigation**—Guides the UAV along a series of waypoints during Lap 1 and Lap 2 within the flight area.
2. **Target area detection & recognition**—Continuously inspects the scene to detect and recognize target areas. During Lap 1, the node searches for the target areas and records their locations, while in Lap 2 it alternates between waypoint following and target tracking.
3. **Precise positioning**—Directs the UAV to position itself directly over the target areas using a vision-based PID controller.

## Autonomous UAV waypoint navigation

Autonomous waypoint navigation enables drones to travel independently along predetermined paths or a series of global coordinates (waypoints) without direct human intervention. Waypoints are typically defined by latitude, longitude, and altitude. The drone's flight control system uses sensor data, such as GPS, altimeters, and gyroscopes to determine its current position and orientation, and then adjusts its flight path, attitude, and actuator outputs to follow the planned trajectory.

Several application programming interfaces (APIs) have been developed for waypoint navigation. One such API, the *Iq_gnc* API (*Intelligent Quads, 2024*) developed by the Intelligent Quads group, provides a variety of functions for controlling UAVs *via* MAVROS. It has been extensively used for testing UAVs in Gazebo-ROS simulation

environments. However, the Python version of the *Iq_gnc* API does not offer a convenient method for implementing waypoint navigation using global geodetic coordinates.

Another popular API among UAV developers is the *DroneKit Python API*. This API provides a powerful and intuitive platform for programming autonomous drones using an object-oriented approach. It simplifies complex tasks, such as autonomous flight—by offering high-level commands that abstract the underlying details of drone control. Its extensive documentation and strong community support further enhance its appeal over the *Iq_gnc* API.

The features of *DroneKit* can be accessed by creating a Vehicle instance using the *connect()* method. This instance grants access to the drone's state, parameters, and sensor data. Methods such as *simple_goto()* can instruct the drone to fly to specified waypoints (defined by geodetic coordinates) in GUIDED Mode. However, one major challenge with this method is the lack of a feedback mechanism to confirm when the target location is reached. Moreover, if the autopilot receives another command before reaching the target, it will immediately execute that command, altering the planned route.

To ensure that no additional commands interfere until the target waypoint is reached, a custom *set_destination()* function was developed. This function employs the *simple_goto()* method from DroneKit Python along with a feedback mechanism to reliably guide the quadcopter to the desired location.

According to the Teknofest mission guidelines (*TEKNOFEST, 2024*), the UAV must complete two laps around the flight area. In the first lap, the UAV follows a sequence of waypoints to locate and record the target areas' coordinates. These coordinates are then dynamically appended to the waypoint list for the second lap. During the second lap, the UAV follows the same route while ensuring that it visits the target areas; at these points, control temporarily switches from Waypoint Navigation Mode to Vision-based Precision Positioning Mode. A behavioral modeling approach is used to facilitate mode switching. Once the UAV completes its tasks at the target areas, control reverts to Waypoint Navigation Mode and the mission resumes. Finally, after both laps are completed, the drone returns to the home position and lands safely.

## Target detection and recognition using OpenCV

This node subscribes to the */webcam/image_raw* topic to receive images from the ROS Gazebo Camera mounted on the Iris drone model in the Gazebo simulation environment. To utilize OpenCV functions, the images must be converted from ROS image format to an array format compatible with OpenCV; this conversion is performed using the CV Bridge package.

Once the images are in the correct format, they are preprocessed using Gaussian Blur for filtering, followed by erosion and dilation to identify contours. Because the target areas are distinguished primarily by color (black for the water intake reservoir and red for the water release area), detection and recognition are performed by converting the image to hue-saturation-value (HSV) format and using the hue value to identify red and blue regions. The contours of these regions are then detected, and if the contour area exceeds a specified threshold, the target area is confirmed.

If the target areas are detected during the first lap, the algorithm publishes a message on the */WDL_WRL_Parameters_topic* to inform other nodes that the target areas have been detected and that their locations are ready to be saved for Lap 2. All relevant flags are transmitted to the ROS Parameter Server, and the locations of the water intake reservoir and water release area are published on the */Water_Reservoir_Location_Topic* and */Water_Discharge_Location_Topic*, respectively.

### Vision-based precise positioning over target areas

This node subscribes to five key topics: */Water_Reservoir_Location_Topic*, which publishes the location of the water intake reservoir; */Water_Discharge_Location_Topic*, which publishes the location of the water release area; *lap_counter_topic*, which indicates the current lap; */WDL_WRL_Parameters_topic*, which transmits flags from the Target Detection and Recognition node; and */mavros/global_position/global*, which publishes the drone's current global position.

After reaching the target areas during the second lap, the drone switches to vision-based precision positioning mode. It receives the current centroid coordinates $(x_t, y_t)$ of the detected target area from the Target Detection and Recognition node and calculates the error $(e_x(t), e_y(t))$ relative to the center of the image window, as shown in Fig. S5. PID controllers in the *xy*-plane direct the drone toward the target center. Once the errors $(e_x(t), e_y(t))$ fall within acceptable bounds, the PID controller in the *z*-plane is activated to initiate descent onto the target. These corrections are converted into appropriate *twist* velocity commands, which are published on the */mavros/setpoint_velocity/cmd_vel* topic. These commands guide the drone to adjust its position precisely above the target (in the x and y directions) and descend while maintaining a safe altitude. The drone hovers at this altitude for a few seconds to perform tasks such as water collection or release. After completing these operations, it ascends back to its previous altitude, and control reverts to *waypoint navigation mode*, resuming the mission.

## EXECUTING THE MISSION

A ROS launch file was created within the appropriate ROS package to start the complete simulation environment including all models and configurations which streamlines the setup process (*Kadri, 2025*).

The DroneKit Python library was configured with the ArduPilot framework (*ArduPilot Development Team, 2025*). The simulation was successfully built on a computer running Ubuntu 20.04 with ROS Noetic installed. The world file (Fig. 6A) and model files (Fig. 6B) were created in *.sdf* format, and all required files and scripts were placed in a dedicated ROS package. After setting up the simulation environment, a basic test was performed to verify that the setup, as shown in Fig. 6C, was working correctly. Mission Planner and MAVProxy were used to send a series of waypoints to the Iris drone model in Gazebo to assess its response. The UAV began executing its mission as soon as its mode was changed to AUTO, and the preliminary test was completed successfully. The motion and trajectory of the UAV were also visualized in Rviz, a well-known ROS visualization tool, as shown in Fig. 6D. The ROS nodes described in the previous section were developed using Python

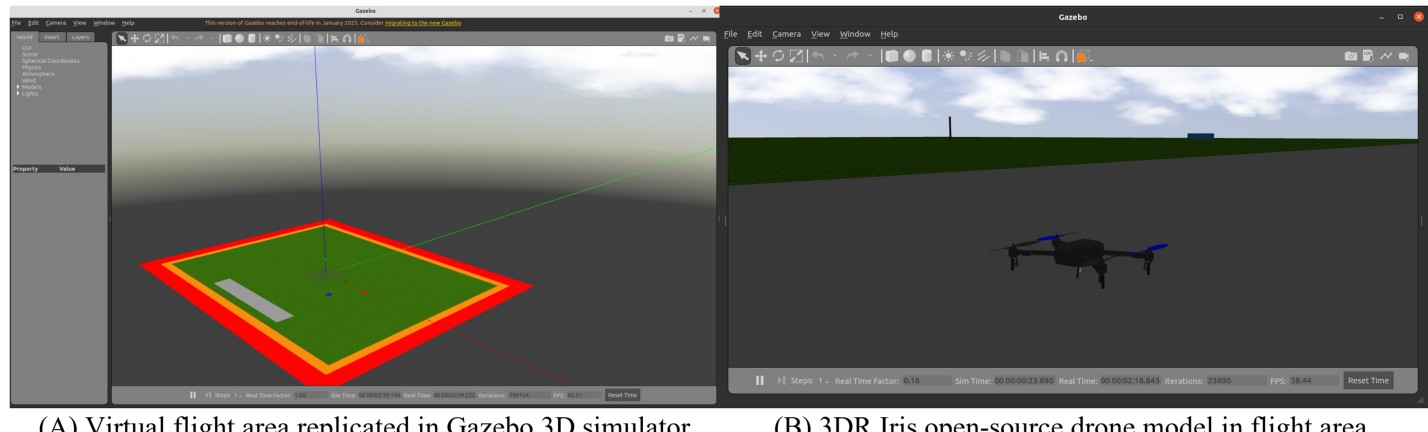

(A) Virtual flight area replicated in Gazebo 3D simulator     (B) 3DR Iris open-source drone model in flight area

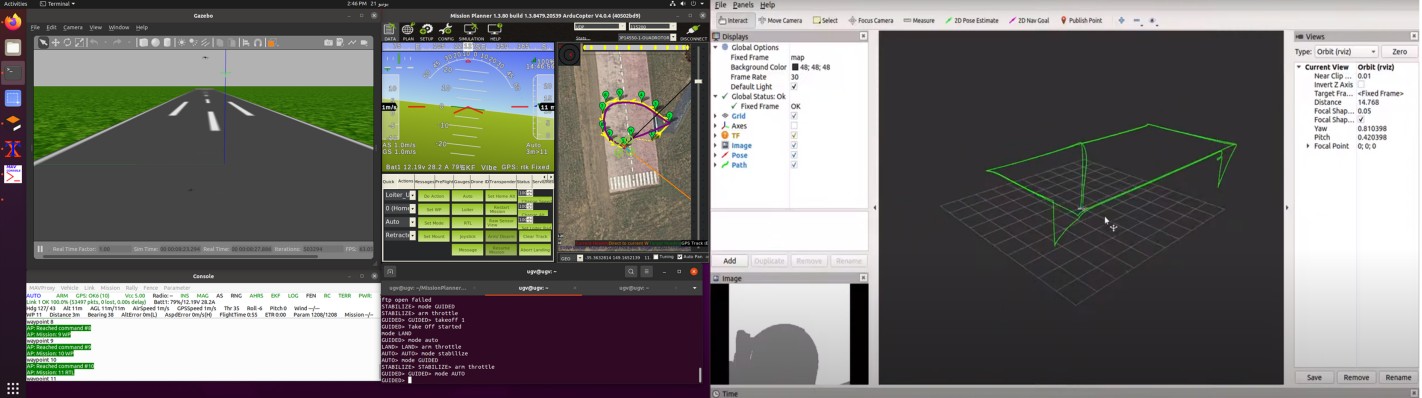

(C) Validating SITL using Mission Planner and MAVProxy     (D) Visualizing UAV trajectory in RViz

**Figure 6** Simulation and testing stages: (A) virtual flight area replicated in Gazebo 3D simulator, (B) 3DR Iris open-source drone model in flight area, (C) SITL testing using mission planner, MAVProxy, and MAVROS, and (D) UAV trajectory visualization in Rviz.

**Table 1 Wind speeds and noise distributions.**

| Condition | Wind speed (m/s) (x, y) | X and Y PID noise $\mathcal{N}(\mu, \sigma^2)$ | Z controller noise $\mathcal{N}(\mu, \sigma^2)$ |
|---|---|---|---|
| Low wind | (1.0, 1.0) | $N(0, 2)$ | $N(0, 0.005)$ |
| Medium wind | (2.5, 2.5) | $N(0, 5)$ | $N(0, 0.025)$ |
| High wind | (5.0, 5.0) | $N(0, 10)$ | $N(0, 0.05)$ |

3.12 and MAVROS. The simulation running in Gazebo 11 took approximately 317 s to complete the mission. The various nodes spawned during the simulation and their corresponding topics are displayed in the ROS graph in Fig. S6. Hovering of the quadcopter over the blue and red tanks is shown in Fig. S7.

## RESULTS AND DISCUSSION

To evaluate the efficacy of the proposed scheme, the quadcopter was tested under various flight conditions, as detailed in Table 1. Sensor noise and wind disturbances were introduced into the system. The robustness of the proposed software architecture under adverse conditions is demonstrated in Figs. 7–10. The results are categorized by different

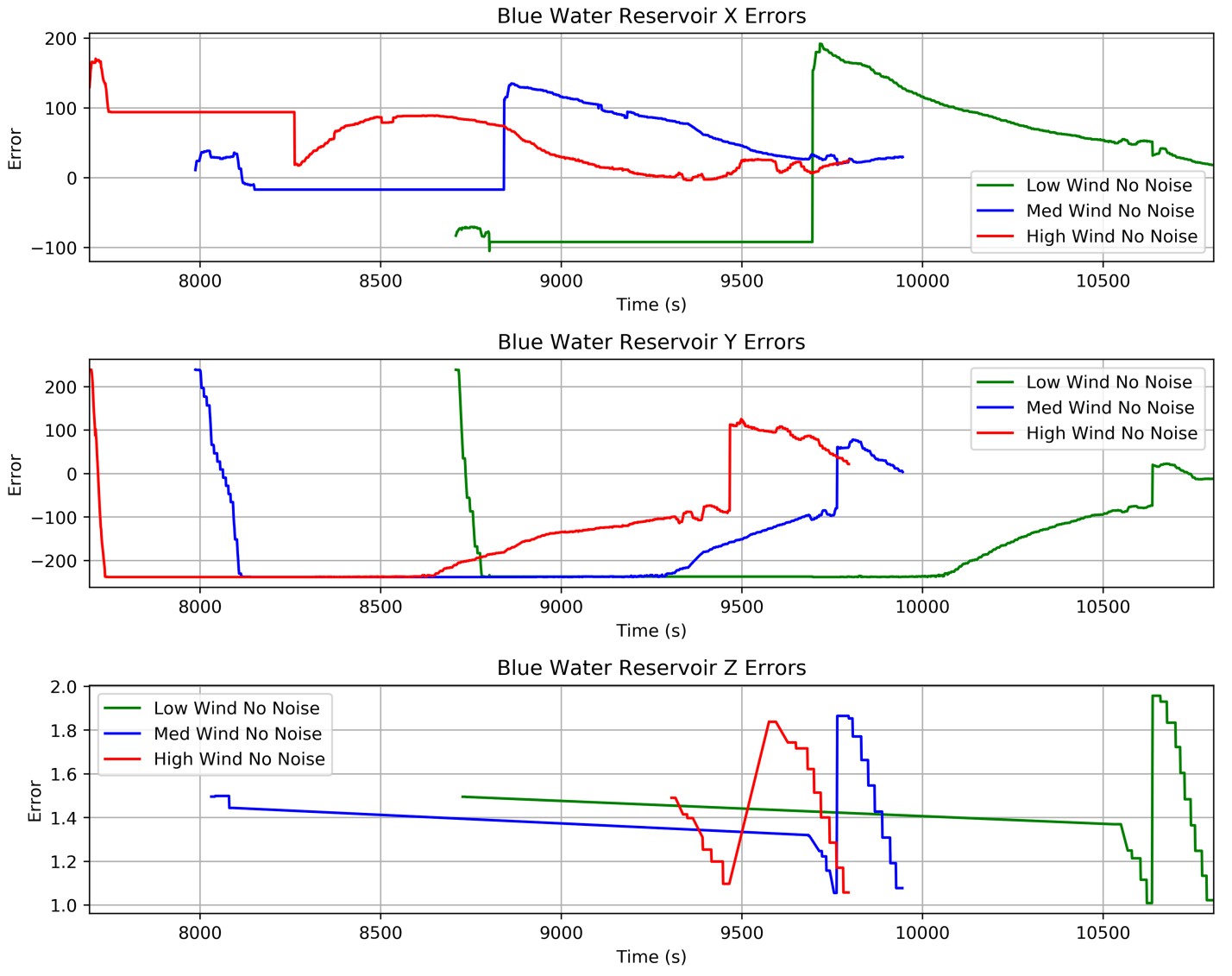

**Figure 7 IBVS PID controller errors recorded over the blue target under effects of different wind conditions and no sensor noise.**

wind conditions and noise levels, providing a comprehensive understanding of the system's performance under various environmental factors.

Figures 7 and 8 illustrate the performance of the x and y PID controllers used by the IBVS system for precise positioning of the drone over the target areas, as well as the performance of the z controller for altitude control under various wind conditions (without sensor noise). Both the x and y controllers work together to position the drone accurately over the target areas. Once the x and y errors reach zero, the z controller activates immediately to descend the drone and then commands an ascent after a short delay to continue the mission. It is also observed that under higher wind speeds, the drone approaches the target areas more quickly; note that the wind was in the same direction as the drone's flight.

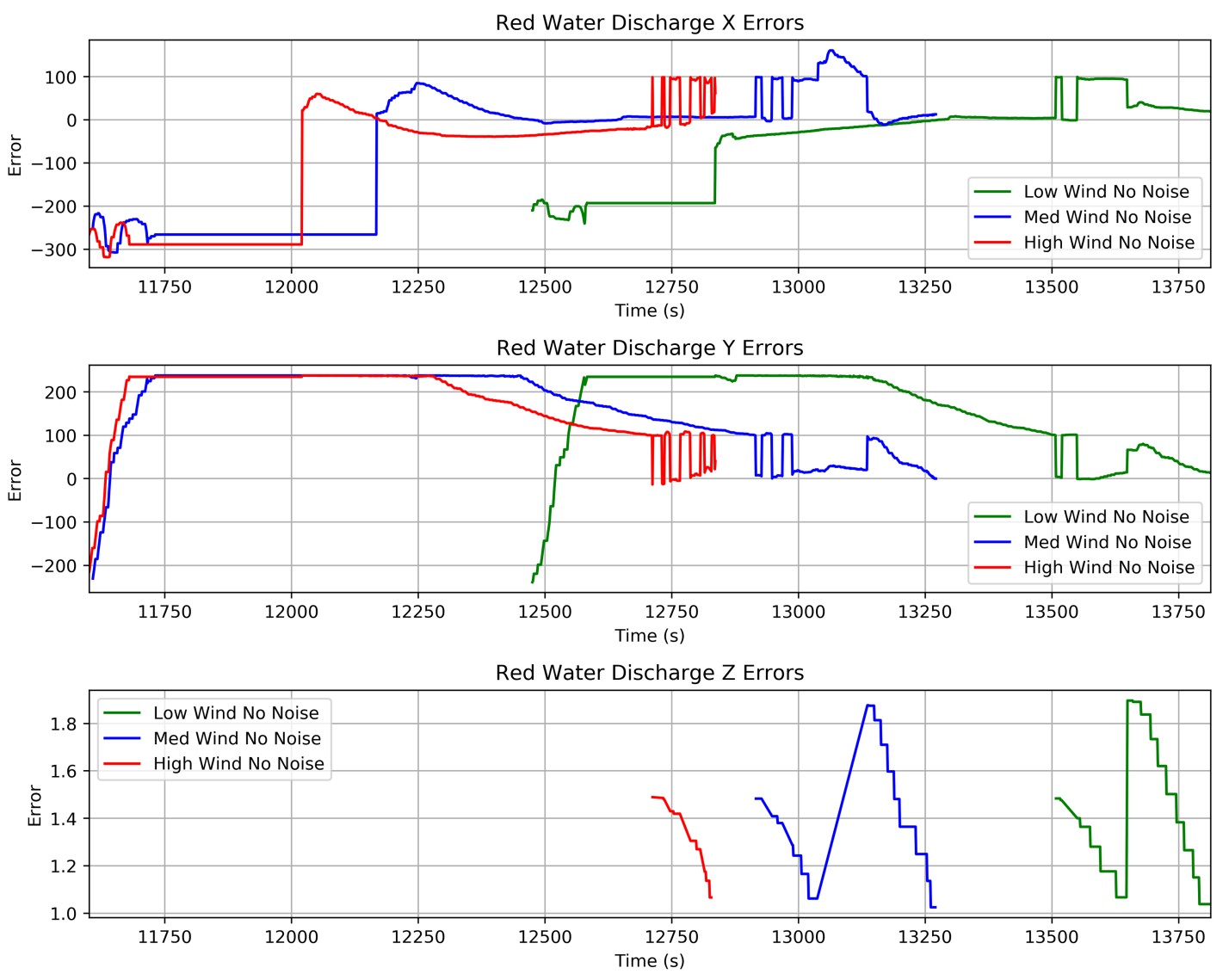

**Figure 8 IBVS PID controller errors recorded over the red target under effects of different wind conditions and no sensor noise.**

In the absence of noise, the system demonstrates stable performance with minimal error fluctuations. However, as wind intensity increases from low to high, the error magnitude also increases especially along the y-axis. The introduction of noise further degrades performance. As seen in Figs. 9 and 10, the error along the y-axis is more pronounced under low wind conditions with high noise compared to scenarios with low noise. This suggests that noise significantly affects the system's ability to maintain precise positioning.

For altitude control along the z-axis, the results indicate relatively stable performance under varying wind and noise conditions. The error in the z-axis remains within a narrow range, demonstrating that the PID controller effectively maintains the desired altitude, though slight variations are observed under high wind conditions. The flow of information

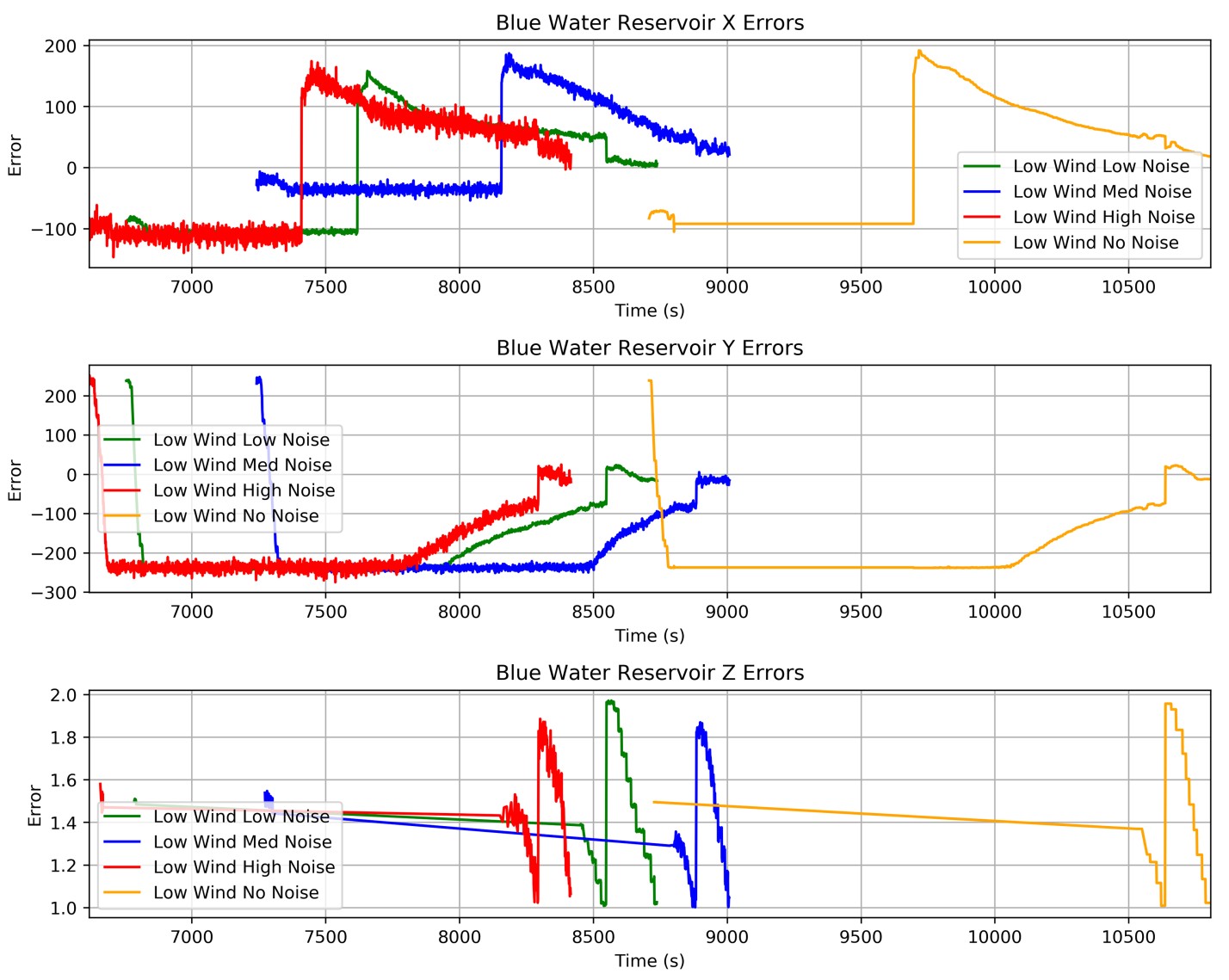

**Figure 9** IBVS PID controller errors recorded over the blue target in low wind condition and different levels of sensor noise.

from the ROS Parameter server to the remote web application through the cloud interface is presented in Fig. S8.

## ROS1 to ROS2 bridge

Figure S9 illustrates the ROS 1 topics being published in ROS 2 *via* the ROS Bridge. Consequently, the proposed comprehensive architecture is compatible with both legacy systems and long-term support (LTS) systems such as ROS 2. The simulation is executed in ROS Noetic, while the bridge publishes all topics to ROS Foxy using subscribers implemented as ROS 2 nodes running on the same machine. These ROS 2 nodes can also be deployed on a networked computer. With ROS 2, the data distribution service (DDS) is

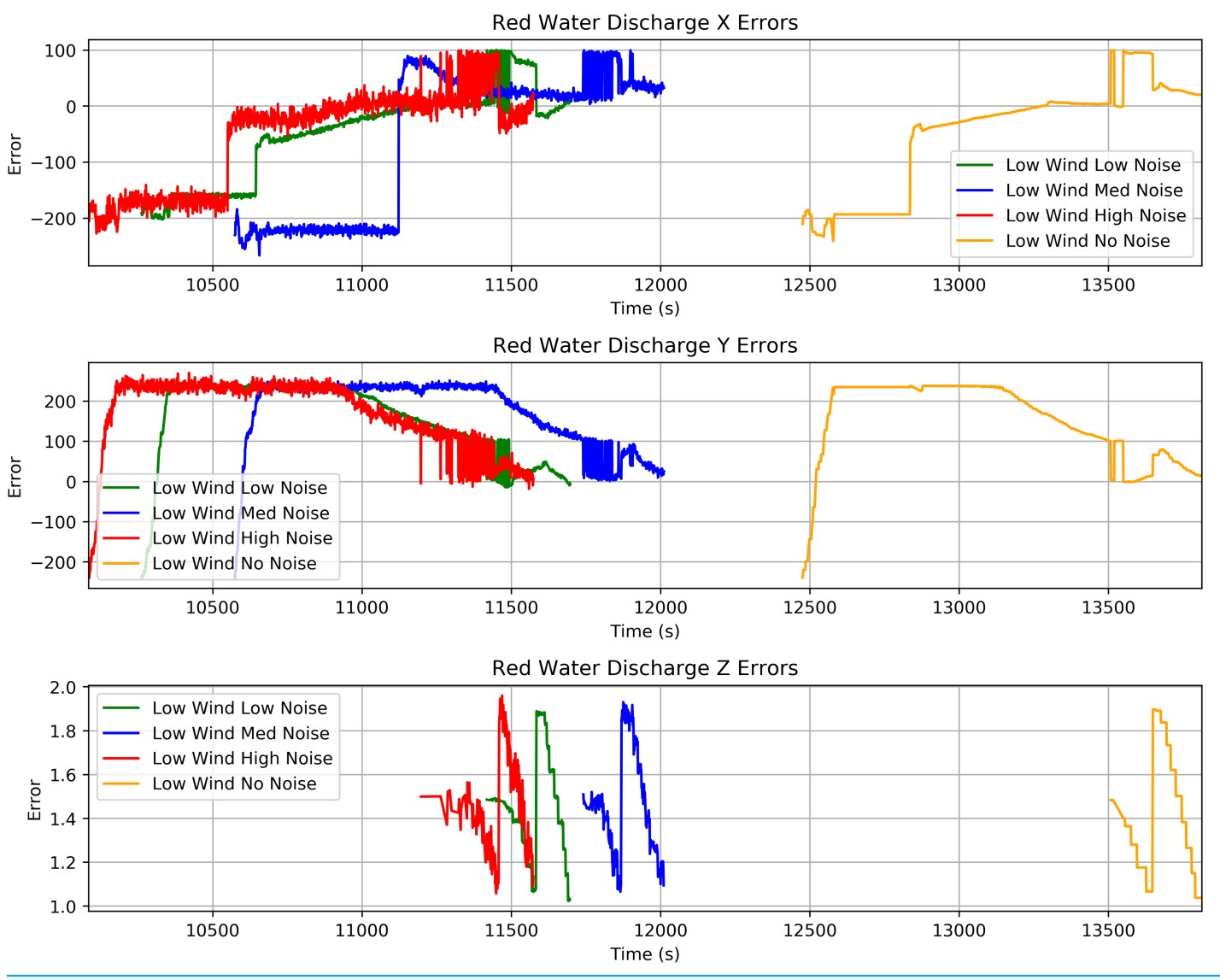

**Figure 10 IBVS PID controller errors recorded over the red target in low wind condition and different levels of sensor noise.**

used for communication, thereby enhancing system robustness, reducing error-proneness, and improving compatibility with cloud-based platforms such as AWS and Google Cloud.

## Statistical analysis

The box plots in Figs. 11 and 12 present a comparative statistical analysis of positional tracking errors in the X, Y, and Z directions during two critical mission phases: approaching the water reservoir (blue target) and descending toward the drop location (red target). Across both phases, an increase in wind intensity from low to high results in noticeable broadening of the interquartile ranges, particularly in the lateral directions (X and Y), reflecting increased uncertainty and variability in position control. During the reservoir approach phase, the X-direction errors show relatively consistent behavior, with

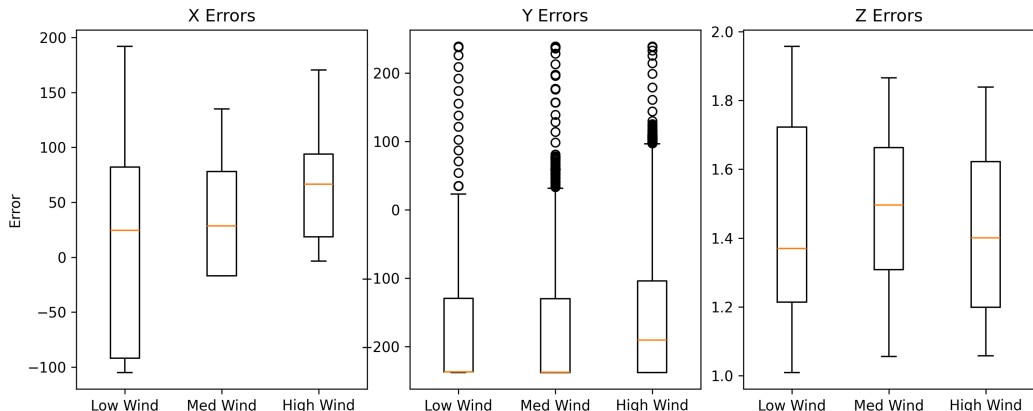

**Figure 11 Box plot for three cases when the quadcopter is approaching the water intake reservoir.**

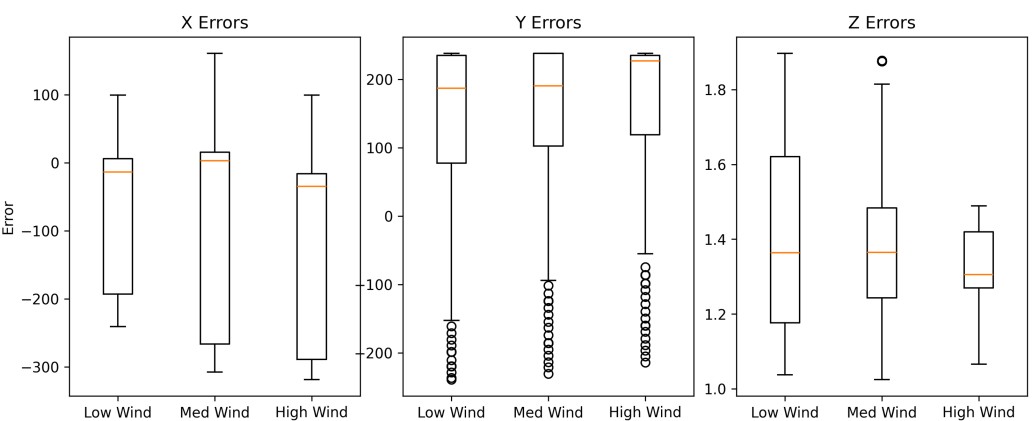

**Figure 12 Box plot for three cases when the quadcopter is approaching the water release area.**

moderate spread and low median error under low wind conditions. However, as wind intensity increases, both the error spread and the median values shift upward, suggesting increased positional drift in the longitudinal direction.

The Y-direction errors exhibit significant sensitivity to wind disturbances, as evidenced by the high density of outliers and increased negative error bias, especially under medium and high wind conditions. This indicates that lateral control is more vulnerable to turbulence when approaching the water source. In contrast, the Z-direction errors remain tightly distributed across all wind scenarios, demonstrating the effectiveness of the altitude controller during descent. The vertical regulation remains consistent and stable, likely due to the dominance of PID-based altitude feedback, which is inherently less affected by horizontal wind components.

**Table 2 List of parameters saved on ROS Parameter server in YAML file required for the mission** (*Reproduced with permission from Raheel, Mehmood & Kadri (2023)*).

| No. | Parameter name | Parameter value | Range |
|---|---|---|---|
| 1 | Lap_Count | 0 | [0–2] |
| 2 | Water_Reservoir_Location_Detected_Lap_02 | 0 | [0–1] |
| 3 | Water_Discharge_Location_Detected_Lap_02 | 0 | [0–1] |
| 4 | Water_Reservoir_Location_Detected_Lap_01 | 0 | [0–1] |
| 5 | Water_Discharge_Location_Detected_Lap_01 | 0 | [0–1] |
| 6 | Water_Sucked_by_Syringes | 0 | [0–1] |
| 7 | Water_Released_by_Syringes | 0 | [0–1] |
| 8 | Water_Reservoir_Location_Latitude | (0.0, 0.0) | (Lat, Lon) |
| 9 | Water_Reservoir_Location_Longitude | (0.0, 0.0) | (Lat, Lon) |
| 10 | Water_Reservoir_Location_Altitude | 0.0 | Height |
| 11 | Water_Discharge_Location_Latitude | (0.0, 0.0) | (Lat, Lon) |
| 12 | Water_Discharge_Location_Longitude | (0.0, 0.0) | (Lat, Lon) |
| 13 | Water_Discharge_Location_Altitude | 0.0 | Height |
| 14 | Water_Discharge_Location_Saved | 0 | [0–1] |
| 15 | Water_Reservoir_Location_Saved | 0 | [0–1] |
| 16 | EXECUTING_WAYPOINT_NAVIGATION | 0 | [0–1] |
| 17 | Current_Waypoint_Index_Lap_01 | 0 | [0–2] |
| 18 | Waypoint_Index_After_which_BLUE_was_detected | 0 | [0–1] |
| 19 | Waypoint_Index_After_which_RED_was_detected | 0 | [0–1] |

In the drop-phase (red target), the controller's performance in X and Y further degrades, with the X-direction errors showing a larger spread and a downward shift in the median under higher wind conditions. The Y-direction again presents a pronounced number of outliers and wider variability, indicating increased difficulty in maintaining lateral stability during terminal descent. Although Z-direction accuracy remains acceptable, it shows slightly more variation compared to the reservoir approach, possibly due to complex airflow near the ground or cumulative disturbances. These findings underscore the robustness of the vertical control channel, while highlighting the need for more advanced or adaptive control strategies in the lateral plane.

## Robustness of the approach

Overall, the results highlight the robustness of the IBVS system and the software architecture in maintaining precise positioning and altitude control under challenging environmental conditions. However, the increased error under high wind and noise conditions suggests that additional measures such as adaptive control strategies or sensor fusion techniques, could further enhance system performance. The stable performance along the z-axis underscores the effectiveness of the PID controller for altitude control. While the implemented controllers demonstrate satisfactory performance, further optimization and the integration of advanced control techniques may improve the system's resilience, ensuring reliable operation under a broader range of conditions. All drone

**Table 3 Gains of the PID Controllers for x and y directions.**

| No. | Parameter name | Parameter value |
|---|---|---|
| 1 | Kp_x | 0.01 |
| 2 | Ki_x | 0.01 |
| 3 | Kd_x | 2 |
| 4 | limit_x | 1 |
| 5 | Kp_y | 0.01 |
| 6 | Ki_y | 0.01 |
| 7 | Kd_y | 2 |
| 8 | limit_y | 1 |

operational data stored on the ROS Parameter Server (as shown in Tables 2 and 3) is transmitted to the cloud *via* a JSON interface. The cloud data is accessible to any remote client, as illustrated in Fig. S8. The PID controller parameters (gains) provided in Table 3 were initially derived using the System Identification Toolbox and later fine-tuned using the Autotune feature of the Ardupilot.

## FUTURE DIRECTIONS

It has been successfully demonstrated that cloud integration with IBVS-based control for UAVs using Ardupilot's SITL is achievable with the proposed architecture. Still, several areas need more research to improve accuracy, safety, and efficiency.

**(1) Stronger sensor fusion for better awareness:** Future systems will use sensor fusion with extended Kalman filters to improve tracking and navigation. Data from IMUs, GPS, LiDAR, and cameras can be combined to reduce noise and deal with sensor failure. These methods can help in rough or low-visibility settings where one sensor is not enough. Adding learning-based filters may also improve tracking when conditions change fast.

**(2) Smarter and flexible control:** Current control uses PID and image-based visual servoing. These work well in simple tasks but may fail in unknown or changing situations. Future work will test adaptive and reinforcement learning-based control. These methods adjust on the fly and improve over time. They can help the UAV deal with wind, changing weight, or small faults. The main problem is that they need more computing power, which drains the battery faster.

**(3) Safer cloud communication:** Since data moves between the UAV and the cloud, it must be safe. Future systems will use tools like intrusion detection, secure key sharing, and blockchain logs. These help detect fake messages, block outside access, and trace system activity. Work will focus on tools that are fast and light, so they do not slow down the UAV or increase the energy use too much.

**(4) Group control of many UAVs:** A single UAV can do small jobs. A group can do more. Future systems will manage many UAVs using one cloud-based control unit. This unit will plan routes, assign tasks, and keep things running if one UAV fails. It will help in wide-area

jobs like farming, search missions, and site inspection. Shared planning also saves time and reduces overlap between UAVs.

These areas offer ways to make UAVs more useful, safe, and ready for large jobs. The goal is to build systems that can act fast, handle rough settings, and work as a team when needed.

## CONCLUSION

In this work, we proposed a comprehensive software architecture for drones and successfully demonstrated its effectiveness in achieving precise positioning of autonomous UAVs over target areas. This was accomplished through the integration of various modules, including autonomous waypoint navigation, target detection and recognition, and vision-based control strategies. These modules were developed within the robot operating system (ROS) framework and integrated with cloud services. A challenging mission was chosen to test the proposed architecture, which was simulated using SITL in Gazebo with the ArduPilot plugin. All modules can be readily configured for real-time execution with minimal effort. The proposed software architecture offers a versatile solution applicable to a wide range of scenarios and real-world applications, paving the way for enhanced precision and reliability in autonomous UAV positioning and enabling advancements across various fields and industries.

### Funding
This work was supported by Prince Sultan University by funding the Article Processing Charges (APC) of this publication. The funders had no role in study design, data collection and analysis, decision to publish, or preparation of the manuscript.

### Grant Disclosures
The following grant information was disclosed by the authors:
Prince Sultan University by funding the Article Processing Charges (APC).

### Competing Interests
The authors declare that they have no competing interests.

### Author Contributions
- Muhammad Bilal Kadri conceived and designed the experiments, performed the experiments, analyzed the data, performed the computation work, prepared figures and/or tables, authored or reviewed drafts of the article, and approved the final draft.

### Data Availability
The code is available at Zenodo:
Muhammad Bilal Kadri. (2025). bilalkadri/Drone-Architecture: Drone Architecture (v1.0.0.0). Zenodo. https://doi.org/10.5281/zenodo.16575404.

The complete source code is available at the following GitHub repository: https://github.com/bilalkadri/Drone-Architecture.git. DOI 10.5281/zenodo.16575404

## Supplemental Information

Supplemental information for this article can be found online at http://dx.doi.org/10.7717/peerj-cs.3238#supplemental-information.

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
