# Peer review of "Enhancing drone autonomy through cloud integration: a comprehensive software architecture for navigation, visual servoing, and control"

_PeerJ Computer Science, doi:10.7717/peerj-cs.3238_

## Round 0.1 · original submission · Major Revisions

Please revise your manuscript addressing all the issues raised by both reviewers, but specially those of reviewer 1.

Reviewer 1 ·

Basic reporting

Clarity and Organization:
The manuscript is organized in a conventional format that begins with an Introduction and clearly moves through problem formulation, description of simulation environments, and experimental results. The narrative is generally unambiguous and professional. Figures—ranging from schematic diagrams (e.g., Figures 1–4 for the control architecture and cloud integration) to detailed mission flowcharts (Figures 6–8) and simulation outputs (Figures 15–18)—are provided to help the reader follow the technical exposition. However, a few typographical and formatting issues (for example, the misspelling “comprehenisive” in the title and occasional inconsistent variable nomenclature) detract from a fully polished presentation. Improving these minor issues would enhance readability, especially for an international audience.

Literature and References: The manuscript cites an extensive set of references spanning original UAV control methods, simulation environments, and cloud interfacing. For instance, the citation of ArduPilot [10] and references pertinent to UAV navigation and sensor fusion [3, 11, 14] underscore that a broad background is considered. Nonetheless, several references (e.g., the DroneKit Python API and the associated code: https://zenodo.org/records/14885778 GitHub repository: https://github.com/bilalkadri/mbk_ardupilot_drone) indicate that the codebase appears to have been last updated approximately three years ago. This raises questions about the currency and, by extension, the potential novelty of the contribution. A more recent update or discussion regarding the evolution of these tools would increase the overall credibility.

Figures and Data Presentation: The quality of figures is generally high, and the diagrams are well labeled. The manuscript includes parameter tables that clearly list gains for PID controllers as well as mission-related variables stored on the ROS Parameter Server. Despite this, some figures (particularly those showing experimental error plots like Figures 19–22) could benefit from a more detailed explanation of the axes, units, and statistical variability. Such enhancements would better align with the PeerJ standards for clear figure descriptions.

Experimental design

Overall Approach: The study presents a simulation-based evaluation of a UAV control architecture integrating cloud connectivity and image-based visual servoing (IBVS). The experimental setup leverages a Software-in-the-Loop (SITL) environment using Gazebo and ArduPilot along with MAVROS and the DroneKit Python API. The three-tier structure covering waypoint navigation, target detection using OpenCV, and precise positioning with visual servoing is adequately detailed. This integrated simulation approach is appropriate for the scope of the paper and addresses key aspects of autonomous navigation.

Simulation Environment and Protocols: The authors describe the simulation setup in detail, including the use of the 3DR Iris drone model, a realistic Gazebo world that replicates a mission scenario inspired by Teknofest 2024, and the inclusion of disturbances such as wind and sensor noise. The methodology for converting image data via CV Bridge to an OpenCV-compatible format is clearly explained, and the successive activation of PID controllers (for x–y positioning followed by altitude control in z) is well documented. However, the experimental design is completely based on simulation with no reported hardware-in-the-loop or field tests. This limitation restricts the ability to generalize the results to real-world conditions.

Parameter Tuning and Replication: While the authors have provided tables listing the gains and parameters used for the PID controllers, the manuscript does not sufficiently describe the process by which these parameters were obtained. More details on the tuning methodology, robustness checks, or sensitivity analyses would greatly enhance reproducibility and allow other researchers to replicate or build on the work. In addition, because the underpinning code repository appears outdated, there is a risk that replication might encounter compatibility issues with current versions of ROS and associated libraries.

Validity of the findings

Effectiveness in Simulation: The results presented in the paper—including error plots for the x, y, and z controllers under various wind speeds and noise levels—indicate that the proposed architecture meets its design goals in the controlled simulation conditions. The simulation data support the claim that a combined IBVS and PID control strategy can achieve precise positioning even under disturbances. For example, the analysis in Figures 19–22 shows that while performance degrades under high wind and elevated noise, altitude control remains relatively robust. This provides reasonable evidence that the designed controllers work as intended within the simulated environment.

Limitations and External Validity: Despite the promising simulation results, the lack of real-world validation remains a significant limitation. The fidelity of the simulation is inherently limited, and factors such as communication latency or sensor uncertainties in real deployments might affect performance. Moreover, while the paper claims robustness under challenging conditions, the quantitative analysis would be more compelling with a deeper statistical treatment—such as error distributions, confidence intervals, or comparative benchmarks against alternative control approaches (e.g., Model Predictive Control)—to substantiate the system’s reliability. Given that some of the underlying software components have not been updated recently, it is also uncertain whether the approach integrates recent advances in adaptive control or AI-based methods for UAV autonomy.

Additional comments

Novelty and State of the Art: The manuscript presents a well-integrated simulation framework that combines cloud-based data logging with traditional IBVS and PID control techniques. Although the technical integration is comprehensive, the novelty is somewhat questionable given that many elements (e.g., ROS-based simulation architectures, PID-controlled waypoint navigation, and the use of standard cloud services) have been established in previous work. The authors should clarify and strengthen the discussion on what new contributions are made relative to state-of-the-art systems, especially in light of the outdated code repository.

Discussion of Cloud Integration: The integration of cloud services for remote visualization is a valuable addition, but the manuscript would benefit from a more critical discussion of the potential challenges—such as security, data latency, and reliability—in real-world deployments. It is recommended that future versions of the manuscript include a discussion on these aspects to provide a more balanced view.

Experimental Enhancements: To further strengthen the paper, the authors might consider including hardware-in-the-loop testing or even flight tests to validate the simulation findings. Additionally, a comparative study with alternative tracking strategies (e.g., using Model Predictive Control or adaptive algorithms) would help position the contribution against existing techniques.

Language and Style: While the overall language is professional, careful proofreading is necessary to address minor typographical errors and ensure consistent terminology throughout the manuscript. This will help in meeting the high standards expected by an international readership.

Future Work and Improvements: The authors are encouraged to discuss potential future directions in greater detail—particularly on integrating more recent developments in sensor fusion, adaptive control, and cybersecurity for cloud-based operations. Explicitly linking these points to the limitations observed in the simulation results will provide a clear roadmap for further research.

Overall, the manuscript is systematically structured and demonstrates a thoughtful integration of UAV control, simulation, and cloud integration techniques. However, issues related to the sole reliance on simulation data, the outdated codebase, and the need for a more in-depth quantitative analysis of robustness call for further revision before acceptance. Addressing these concerns would greatly increase the contribution’s impact and relevance in the evolving field of autonomous drone navigation.

Reviewer 2 ·

Basic reporting

Figures should include clear citations and detailed proofs. The clarity of the images included in the research article is missing. The research paper lacks formal theorems or detailed mathematical proofs. Citations are required for the images included in the research article. It aligns well with current trends in UAV research, including cloud integration, real-time control, and vision-based navigation.

Experimental design

The research question is relevant and meaningful, addressing the challenge of enhancing drone autonomy for real-world missions requiring precise positioning and robust control. The manuscript explicitly states the knowledge gap: existing simulation architectures mostly focus on navigation but do not comprehensively cover target identification and cloud integration. It highlights the limitations of traditional GPS-based navigation for small target areas and proposes IBVS as a promising solution. The research conforms to prevailing ethical standards in UAV software development and simulation studies. The article mentions adherence to data-sharing policies by providing raw data and source code openly via GitHub, supporting transparency and reproducibility.

Validity of the findings

The manuscript does not explicitly assess the broader impact or novelty of the proposed software architecture in the findings section. However, the work implicitly demonstrates novelty by: Integrating cloud-enabled autonomous navigation with image-based visual servoing (IBVS) and control within a unified ROS/Gazebo simulation. Addressing the gap in existing simulation architectures that typically lack combined target identification and cloud integration.

---

## Round 0.2 · accepted · Accept

Since the revised version has addressed most of the remarks made by the reviewers, it can be accepted. However, I ask the authors to better justify/explain the adjectives in section 1.1 :
"...[unified, modular and cohesive (?)] control framework that can be [feasibly deployed on low-cost (?) UAV platforms].

Reviewer 2 ·

Basic reporting

No comment

Experimental design

No comment

Validity of the findings

No comment

Additional comments

The updated version fits all the criteria and is now ready for publishing.